# Genetically Targeted Clinical Trials in Parkinson’s Disease: Learning from the Successes Made in Oncology

**DOI:** 10.3390/genes12101529

**Published:** 2021-09-28

**Authors:** Magnus Sjögren, Henri J. Huttunen, Per Svenningsson, Håkan Widner

**Affiliations:** 1Herantis Pharma Plc, Bertel Jungin Aukio 1, 02600 Espoo, Finland; henri.huttunen@herantis.com; 2Department Health Sciences, Umeå University, 907 36 Umeå, Sweden; 3Department of Neurology, Karolinska University Hospital, 171 77 Stockholm, Sweden; per.svenningsson@ki.se; 4Department of Neurology, Lund University Hospital, 221 00 Lund, Sweden; hakan.widner@med.lu.se

**Keywords:** Parkinson’s disease, neurodegenerative disorders, genetic targets, clinical trials, oncology, precision medicine

## Abstract

Clinical trials in neurodegenerative disorders have been associated with high rate of failures, while in oncology, the implementation of precision medicine and focus on genetically defined subtypes of disease and targets for drug development have seen an unprecedented success. With more than 20 genes associated with Parkinson’s disease (PD), most of which are highly penetrant and often cause early onset or atypical signs and symptoms, and an increasing understanding of the associated pathophysiology culminating in dopaminergic neurodegeneration, applying the technologies and designs into the field of neurodegeneration seems a logical step. This review describes some of the methods used in oncology clinical trials and some attempts in Parkinson’s disease and the potential of further implementing genetics, biomarkers and smart clinical trial designs in this disease area.

## 1. Introduction

Early February this year, it was announced that the Phase II SPARK study of cinpanemab, a candidate drug for Parkinson’s disease (PD), missed its primary and secondary endpoints. The purpose of the SPARK study was to investigate the effect of cinpanemab, an anti-α synuclein (syn) monoclonal antibody, on PD symptoms and compare the effect with placebo in patients with idiopathic PD [1]. This is only one of a number of failed clinical trials in PD over the past several years going outside of the classical dopaminergic class of drugs. In fact, the last 15 years has seen a fair share of failures in late-stage development of medicines not only for PD but for neurodegenerative disorders in general. The associated costs have been significant leading to the decision by several pharmaceutical companies to withdraw from this disease target area until further progress in basic and applied science has been made. 

Fortunately, substantial progress in the knowledge of how PD and other neurodegenerative disorders develop with findings both in molecular biology, genetics, and clinical factors, have been made. Steps have been taken in neuroscience as well as in other disease areas, to attempt to catch up with what is seen in oncology, currently the most advanced disease area in terms of progress in innovation of medical treatments. Here, in the field of oncology, the paralleled increase in understanding of disease biology such as how genetics links to pathophysiology, as well as how to apply this knowledge in the development of novel medicines, has been nothing less than remarkable with several new medicines for treatment of cancer being launched over the past few years alone [2]. Attempting to mimic this in neurological disorders is obvious: However, there is a shortage of clinical trials implementing the oncology approach in terms of study designs and the concrete application of tools and knowledge in the understanding of disease biology in clinical trials. Recently, the PASADENA trial study of an α-syn antibody in PD suggests that some steps in the direction of implementing biomarkers for decisions making have been made [3], using these as gateways before escalating the trial to expose larger number of individuals for a longer period of time.

## 2. Disease Biology in PD

Taking a step back to disease biology, the PD field has evolved substantially in the last three decades to lay the ground for precision medicine. In the mid-1990s, the connection between PD and underlying genetic mutations was established [4,5,6], and today the notion seems substantiated that varying degrees of the interplay between genomic predispositions, aging and cellular stressors impose a clear risk for developing PD [7]. Previous studies on risk factors have described vascular insults to the brain, repeated head trauma, exposure to neuroleptic drugs, to pesticides, and to manganese related to a risk increase of developing PD [8,9,10]. Furthermore, as in Alzheimer’s disease, advancing age is associated with an increase in the amount of cellular stressors which in PD has its epicenter within the substantia nigra, making the neurons vulnerable and less ready to respond to additional insults [11,12]. This plethora of interactions between genes and the environment puts a challenge on the development of a single treatment for PD [13].

In genetic research, mutations in more than 20 genes have been associated with PD, most of which are highly penetrant and often cause early onset or atypical signs and symptoms. The MDSGene database currently contains data on 2463 different genetic variants in 8625 movement disorder patients extracted from 1559 publications such as case and family-based studies and mutation screens (http://www.mdsgene.org/ accessed on 1 May 2021). About 5–10% of patients have familial PD with often causative mutations in autosomal-dominant (AD) genes–SNCA, LRRK2, and VPS35 and autosomal recessive (AR) genes–PINK1, DJ-1, and Parkin (Figure 1). The use of whole-exome sequencing has identified AR DNAJC6 mutations in predominantly atypical PD i.e., Lewy-body dementia, corticobasal degeneration, progressive supranuclear palsy and multiple system atrophy, as well as in typical PD [14]. In addition, to date, genome-wide association studies have identified >100 independent risk-associated variants, whereof most have been identified in patients of European ancestry and little of the genetics of PD in other populations is known. Still, a limited understanding of the biological functions of the risk alleles that have been described, although many PD risk variants appear to be in close proximity to known PD genes and lysosomal-related genes.

While accumulation of aggregates α-synuclein in Lewy bodies and neurites is considered a neuropathological hallmark of PD, the underlying reasons resulting in this form of proteostasis failure in idiopathic forms of PD remains poorly understood. Dysfunction of various cellular pathways have been associated with PD pathogenesis, including defective protein degradation (ubiquitin-proteasome system (UPS) and autophagy-lysosomal pathway (ALP)), chronic endoplasmic reticulum (ER) stress, intracellular trafficking failures, mitochondrial dysfunction, oxidative stress, and calcium mishandling [15]. Many of the known PD risk genes are functionally associated to these cellular systems providing insight on how genetic predispositions and α-synuclein misfolding/aggregation may be linked to degeneration of nigrostriatal dopamine neurons [16]. For example, PD-associated mutations of GBA reduce the glucocerebrosidase enzyme activity, which affects the accumulation and clearance of α-synuclein and lysosomal homeostasis, exacerbating ER stress and promoting mitochondrial dysfunction [17]. Other examples are parkin (PARK2), an E3 ubiquitin ligase, which supports removal of damaged mitochondria and suppresses accumulation of α-synuclein aggregates [18], and PINK1, a mitochondrial serine/threonine-protein kinase, that can suppress α-synuclein-mediated degeneration of nigral dopamine neurons [19]. LRRK2 has been functionally associated with regulation of endolysosomal trafficking and autophagy in neurons [20] but is also expressed in immune cells indicating its pathophysiological roles in PD may be multilayered [21]. Importantly, human iPSC studies of a wide range of genetic subtypes of PD have highlighted converging molecular and cellular pathways, such mitochondrial dysfunction and impaired proteostasis, across genetic subgroups [13].

Interestingly, recent studies have suggested that PD may be not only a proteinopathy but also a lipidopathy [22]. Around 40% of CSF alfa-Syn is found in lipoprotein particles [23]. Finally, as in most chronic diseases, inflammation contributes to the pathogenesis of PD. Recently, it was proposed that PD pathogenesis can be divided into three temporal phases, and that genetic and environmental risk factors of PD could be categorized in triggers, facilitators, and aggravators affecting different temporal phases [24]. Due to this etiopathological heterogeneity, PD is now increasingly considered a complex disorder that encompasses various clinical, epidemiological, and genetic subtypes. Precision medicine approach would thus be well suited for disease modification in PD, with an aim to target a defined PD subtype with a therapeutic accompanied with specific disease-related genetic, molecular, and pathological biomarkers.

## 3. Bringing Knowledge of Disease Biology into Clinical Trials

One of the obvious challenges with clinical trials for a disease modification label in PD is the duration of the trials required for demonstrating a statistically significant difference from standard of care. An annual 4.7 point change in UPDRS [25] assuming a continuous and linear decline, would require very lengthy trials with a readout after at least 2–3 years. On the other hand, should there be a non-linear decline in UPDRS, even longer trials may be needed [26]. This would especially be the case if very early-stage PD patients were enrolled having a very slowly progressive disorder. However, taking lessons designing clinical trials in oncology medicine may improve the situation for neurodegenerative disorders. Frequently, the starting point for designing trials in oncology is a clear notion of the disease mechanisms involved. This change has evolved following the perception of tumors as genetic diseases as opposed to tissue-dependent processes, which has accelerated the time from discoveries to approvals in the oncology therapeutic area. Translating this to PD, this would imply reclassifying the disorder into subgroups based upon genotype, an approach that seem to be taking shape during the last few years.

In PD, one of the most important risk factors is mutations of the glucocerebrosidase gene, GBA1 (OMIM 606463), which is associated with a decrease in the activity of the β-glucocerebrosidase (GCase) enzyme. Ambroxol, a medicine that is used in respiratory disorders to break up phlegm, has been found to increase GCase activity, and both in vitro and in vivo studies have found that ambroxol also reduces α-synuclein levels. Thereby, using ambroxol as a treatment in PD may mechanistically interfere with the pathophysiology of the disease and lead to a reduction of α-synuclein levels in the short term and a disease modification in the long term. In 2020, a non-randomized, non-controlled study was presented in *JAMA Neurology*, which had exposed 17 patients with PD to ambroxol. They found that ambroxol crossed the blood-brain barrier and bound to the β-glucocerebrosidase enzyme, and it increased β-glucocerebrosidase enzyme protein levels and cerebrospinal fluid α-synuclein levels in patients both with and without glucocerebrosidase gene mutations [27]. The study was focused on understanding disease mechanisms and not on clinical efficacy, and with the inclusion of a biomarker such as CSF-α-synuclein, demonstrating an increase serves as an indication of a biological effect that increases the confidence that ambroxol in the long term may modification disease progression. Currently, there are three studies with ambroxol as registered in clinicaltrials.gov, whereof the above mentioned is one which is completed and two additional are still ongoing.

## 4. Challenges and Opportunities of Genetically Targeted Clinical Trials in PD

An approach that is currently pursued in oncology trials is the multiphase optimization strategy (MOST) and the sequential multiphase adaptive randomized trials (SMART) [28]. MOST incorporates the standard randomized controlled trial (RCT), but before the RCT is undertaken also includes a principled method for identifying which components are active in an intervention, and which doses of each component lead to the best outcomes. Translating this to a trial in a neurodegenerative disorder, this would imply making changes in dose and treatment arms based upon biomarker responses along the running of the trial. The SMART approach may innovate clinical trials when different treatments are tried in the same trial, either administered as combinations or sequentially over time.

The MOST and SMART approaches may not be implemented in trials in neurodegenerative disorders in the very near future. Currently, the enrichment methods in PD trials have focused on a certain type of set of symptoms, e.g., REM sleep disorder, or a specific genotype. Options for stratifying in disease modifying studies applying enrichment may make use of different polymorphisms having dissimilar penetrance enabling gradients of biological impact adding to the exposure dimension to gain a deeper understanding on mechanistic aspects of the disorders. The enrichment approach, however, has some potential drawbacks since usually, these populations are both hard to find and sometimes suffer from more aggressive disease progression as with GBA mutations in PD, which may provide opportunities for earlier read-outs, while at the same time pose additional challenges being extraordinarily hard to treat. Early engagement with patient organizations and ethical committees to enable initiation of innovative treatments at an earlier stage. In addition, should many companies put their main focus on patients with mutations this could lead to a competitive situation with a risk of diminishing access and impairing innovation.

Despite the obvious opportunities with novel designs applied also in clinical trials in neurodegenerative disorders, and with bringing increased disease mechanistic knowledge into the selection of molecular targets to treat, there are also recent failures applying this specific approach that may dampen enthusiasm to pursue the oncology approach in neuroscience. The failure of the venglustat, a glucosylceramide synthase inhibitor, to slow down disease progression in PD patients with GBA mutations is a most recent example. The analysis of results obtained with venglustat is currently ongoing and no data has been published. Questions to be discussed are at which stage of the disease PD patients with GBA mutations would be amenable to treatment with a compound targeting a GBA-based mechanism, as well as whether there may be additional pathophysiological aspects to be discovered that add to the development of the disease in PD patients with a GBA mutation. Thus, there is no reason to abandon the hopes for developing effective treatments of neurodegenerative disorder when attempting to follow the methods applied with such a success in the field of oncology.

Fortunately, a lot of knowledge will be gained from the more than 140 human studies ongoing in PD, which includes thirteen α-synuclein targeting treatments, e.g., antibodies, vaccines, gene therapies, and small-molecule compounds, and four potential GBA targeting treatments, one gene therapy, and three small-molecule compounds. In addition, in August 2020, a San Francisco-based biotech company announced a $1-billion deal to develop therapies targeting the LRRK2 protein, an overactive gene in PD, which clearly underscores that the PD field is pursuing the oncology approach, adding another genetic target to the growing list, increasing the hopes for patients with PD that there may be disease modifying remedies in the next decades to come, at least for subsets of patients with PD.

For clinical trials in genetically defined subpopulations of PD, variants of relatively high prevalence, high effect size and well-established functional connections to disease biology would be the most attractive targets. The glycine to serine substitution (G2019S), located within the protein kinase domain of LRRK2 protein, is the most common mutation causing PD. It was estimated by the international LRRK2 consortium to represent 1% of sporadic and 4% of familial PD patients worldwide [29]. However, LRRK2(G2019S) is not fully penetrant and not all carriers develop Lewy body pathology making this mutation not a perfect target for precision medicine trials.

More than 300 variants of the glucocerebrosidase (GBA) gene have been associated with Gaucher’s disease, a lysosomal storage disorder. Years ago, it was noted that there is an increased frequency of parkinsonism among patients with Gaucher disease and today it is known that GBA mutations represent the single largest risk factor for development of idiopathic PD [30].

With both LRRK2(G2019S) and GBA mutations, pathological pathways of interest include changes in lysosomal-autophagy pathway function, endoplasmic reticulum (ER) stress, and α-synuclein aggregation.

Increasing interest towards clinical trials focusing on genetic subtypes of PD may pose new practical challenges. First of all, patients carrying risk variants are scarce, often not aware of their genotype and may present with a clinically typical PD. For example, LRRK2(G2019S) and GBA(N409S) carriers often have a clinically indistinguishable disease from idiopathic PD with resting tremor, bradykinesia, rigidity, and postural instability. Secondly, disease modification trials typically have long durations (years) and there may be significant competition for eligible patients carrying a specific genotype. However, since genotyping or sequencing large numbers of PD patients is expensive and slow, genetically focused clinical trials may need to implement special patient recruitment strategies that consider geography, ethnicity, and ways to enrich potential carriers for genotyping based on their disease characteristics.

Different genetic variants are known to be represented some ethnic groups and sometimes in particular phenotypes suggesting that trials could be geographically targeted in specific areas (Figure 2). Moreover, clinical phenotypes, age of onset, and disease progression rate data could be used to enrich patients for genotyping. For example, the frequency of LRRK2(G2019S) is 30–41% in familial PD, especially in Ashkenazi Jews (up to 18%; [31]) and 30–39% in apparently sporadic PD in North Africa [32]. Within healthy controls, Moroccan Berbers appear to have the highest LRRK2(G2019S) carrier frequency at 3.3%. Due to historical reasons, there is a large population of Moroccan immigrants (>1 million) and their descendants in France making this country a potentially attractive region for clinical trials focused on LRRK2(G2019S) carriers. On the other hand, the most frequent GBA mutation, N409S (previously known as N370S), is rather common in patients from Europe, America, and the Middle East, while it is rarely observed in Chinese or Japanese populations. Among Ashkenazi Jews, one in 14–18 individuals harbor GBA mutations, and N370S accounts for 70% of the mutant alleles [33]. As GBA(N409S) is commonly found among people with Ashkenazi Jew origin, trials aiming to recruit GBA(N409S) carriers should focus on areas with high frequency of people with Ashkenazi Jewish origin (e.g., >2 million Jews live in New York state, most with Ashkenazi origin).

Continuous aggregation of patient data in registries could offer an efficient way of enriching well-phenotyped patients for genotyping. This type of approach has been developed for Alzheimer’s disease [34] and would certainly offer a useful aid for accelerated patient screening for precision-medicine clinical trials in PD. Consumer genomics tools such as home test kits, fast genotyping/sequencing services, and existing databases with genetic profiling data could also significantly facilitate implementation of precision medicine trials in PD. New concepts and service providers are also appearing in the field with the aim of creating trial-ready cohorts to accelerate clinical development of novel therapies.

## 5. Conclusions

The future of PD clinical trials will be patient-centric and data driven. Precision medicine approach pioneered in oncology offers many lessons to be adapted to clinical trials in neurodegenerative disorders. For PD, there are some early attempts of implementing early read-outs and enriching on familial PD, while the innovative approaches characteristic of the oncology field, and precision medicine, is yet to be implemented.

## Figures and Tables

**Figure 1 genes-12-01529-f001:**
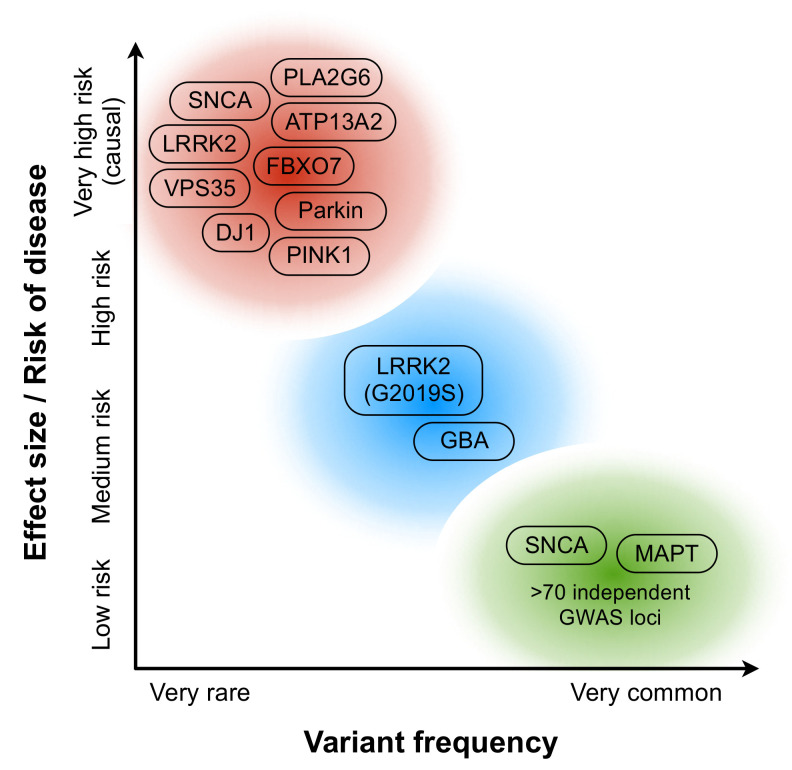
Genetic landscape of Parkinson’s disease. In the continuum of variants of different effect strengths and allele frequencies, the red bubble includes variants and mutations of high effect size/risk of disease, and the green bubble includes common polymorphisms with small effect size. Due to their relatively high prevalence and effect size, LRRK2 (G2019S) and some GBA variants (such as N409S and L483P; blue bubble) are of special interest for clinical trials focusing on genetic subtypes of PD.

**Figure 2 genes-12-01529-f002:**
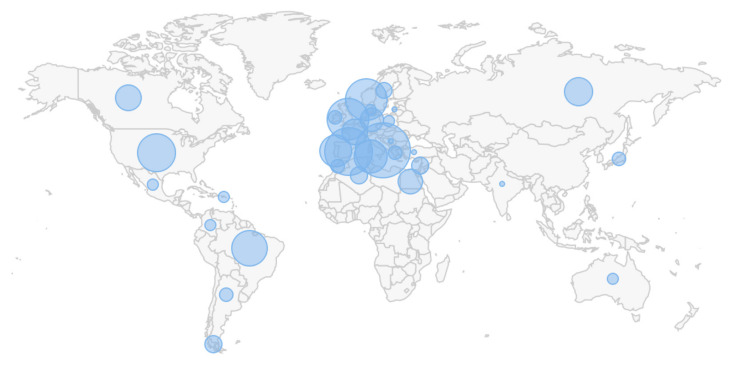
World map showing locations of known LRRK2(G2019S) mutant families based on MDSGene database (http://www.mdsgene.org/; 1 May 2021). North Africa and Western Europe show high frequency of this mutation indicating that clinical trials focusing on this genetic subtype of PD may benefit from focusing on these regions.

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
