# Peer review of "Genetically Targeted Clinical Trials in Parkinson’s Disease: Learning from the Successes Made in Oncology"

_genes, 2021, doi:10.3390/genes12101529_

Round 1

Reviewer 1 Report

In this manuscript, the authors clearly described  targeted clinical trials in PD.

Author Response

We thank this Peer reviewer for comments and feedback

Reviewer 2 Report

The manuscript by Magnus Sjögren and colleagues provides a summary of the current clinical trials in Parkinson’s disease (PD), and new strategies to implement here, based on methods used in oncology, in order to develop effective treatments for this neurodegenerative disease.

PD is considered a complex disorder, which includes different clinical, epidemiologic and genetic subtypes. This condition makes difficult to find an effective treatment that reaches a definitive cure. Here, the authors point out a new therapeutic strategy, focusing on disease molecular mechanisms rather than on clinical efficacy.

The review is well structured and the main issues well presented. However, the document does not add significant value to the field of clinical trials in PD.

Author Response

We thank this peer reviewer for comments and feedback. Concerning this peer reviewers personal opinion, we indeed think that this review is providing a much needed novel perspective on clinical trials in parkinsons disease.

Reviewer 3 Report

Sjogren et al have written a short commentary piece on learning from successes made in the field of Oncology with regards to the realization of clinical trials for treating PD. This is a relevant discussion point and of interest.

Here are my main critical comments and some questions that the authors could consider:

  • In the abstract, the comparison to oncology trials is justified. However, cancer research has a major advantage over PD research in that cell and tissue specificity is much easier to model at the bench. Later the authors refer to using knowledge from basic research to drive clinical trials more smoothly (which is a good point) but on first glance it appears that the authors overlook the fact that brain researchers have been struggling for many years to understand the science behind the specific vulnerability of the dopaminergic neurons in the substantia nigra. Cancer researchers can work directly on the tumour cells, PD researchers only relatively recently were able to work in human dopaminergic neurons and even now these cells (if derived from stem cells) are not ‘aged’. Some of these points are mentioned later in the text but it would be a more balanced abstract summary to recognise this major head start in the oncology field.
  • Paragraph ending line 30. We should not forget that some positives also came from these trials. We learnt that the disease starts much earlier that we previously thought. It would be a shame for all the participants of these trials and all the research that took place to not recognise at least one positive outcome. Again for balance.
  • Sentence starting line 42. Can the authors clarify what they want to say? It is a confusing sentence. Perhaps it could be made more precise? The trial being referred to is using aSYN antibodies and what is it about this study that suggests that biomarkers for decision making was done exactly? Which biomarker? What makes this trial a good example?
  • Line 54. Disease Biology in PD. Sentence starting Furthermore, as in Alzheimer’s disease. Please re-write. It could be understood that in Alzheimer’s disease there is an increase of cellular stresses within the substantia nigra also.
  • Line 87. Is there a reference missing?
  • Large paragraph starting line 75. Is there any evidence that PD risk gene biomarkers are reflected in sporadic PD? Or as a subgroup of sporadic PD. Could the authors refer to a study where a biomarker such as aSYN seeding is seen in genetic risk, genetic PD and in a proportion of sporadic PD (with significance)? It might be an important point as there is some discussion over whether monogenic PD might be distinct from sporadic PD, risk gene PD. Also line 82, clarify ‘known PD risk genes’ with a few examples.
  • Line 113. It is correct that viewing tumours as a genetic disease has helped. And considering the heterogeneity of PD, it could help also. But most cases of PD are sporadic and DNA from those patients often not routinely screened like tumour samples are. Perhaps I misunderstood, but are the authors suggesting that we should identify common genetic (risk?) signatures in PD to be able to have better trials and treatments?

Minor points, English language and typos

  • Line 43. Typo. Decisions making should be decision making.
  • Line 87. ‘alfa’ should be ‘alpha’ to be consistent with first use on page 1.
  • Comment Figure 1/main text. Would it be relevant to mention zygosity? Heterozygous, compound heterozygous cases in PD?

Author Response

We thank this peer reviewer for valuable comments and feedback. Below are our responses to his/her comments:

  • In the abstract, the comparison to oncology trials is justified. However, cancer research has a major advantage over PD research in that cell and tissue specificity is much easier to model at the bench. Later the authors refer to using knowledge from basic research to drive clinical trials more smoothly (which is a good point) but on first glance it appears that the authors overlook the fact that brain researchers have been struggling for many years to understand the science behind the specific vulnerability of the dopaminergic neurons in the substantia nigra. Cancer researchers can work directly on the tumour cells, PD researchers only relatively recently were able to work in human dopaminergic neurons and even now these cells (if derived from stem cells) are not ‘aged’. Some of these points are mentioned later in the text but it would be a more balanced abstract summary to recognise this major head start in the oncology field.
    • we agree with this Peer and have added this difference to oncology trials, while stressing the opportunity that genetics, CSF and blood biomarkers may provide in bringing basic research closer to the clinic. The following has been added to the abstract:
    • "...and an increasing understanding of the associated pathophysiology, which endpoint includes dopaminergic neurodegeneration,..."
  • Paragraph ending line 30. We should not forget that some positives also came from these trials. We learnt that the disease starts much earlier that we previously thought. It would be a shame for all the participants of these trials and all the research that took place to not recognise at least one positive outcome. Again for balance.
    • Since this is one of our main points, also reflecting the real situation with a lot of failed late stage clinical trials, it would be contradictive to  balance this sentence. The whole purpose is to explore novel ways into treating the neruodegeneration in PD, which the dopaminerigc drugs fail to do. We have added a sentence to clarify this, which reads: "going outside of the classical dopaminergic class of drugs."
  • Sentence starting line 42. Can the authors clarify what they want to say? It is a confusing sentence. Perhaps it could be made more precise? The trial being referred to is using aSYN antibodies and what is it about this study that suggests that biomarkers for decision making was done exactly? Which biomarker? What makes this trial a good example?
    • we have added some words to this sentence for clarification: "Here, in the field of oncology, the paralleled increase in understanding of disease biology such as how genetics links to pathophysiology, as well as how to apply this knowledge in the development of novel medicines, has been nothing less than remarkable with several new medicines for treatment of cancer being launched over the past few years alone [2]."
  • Line 54. Disease Biology in PD. Sentence starting Furthermore, as in Alzheimer’s disease. Please re-write. It could be understood that in Alzheimer’s disease there is an increase of cellular stresses within the substantia nigra also.
    • we agree with this peer and have edited the sentence. It now reads: "Furthermore, as in Alzheimer’s disease, advancing age is associated with an increase in the amount of cellular stressors which in PD has its epicenter within the substantia nigra, making the neurons vulnerable and less ready to respond to additional insults [11, 12]."
  • Line 87. Is there a reference missing?
    • We believe we have included a reference that covers all these aspects mentioned.
  • Large paragraph starting line 75. Is there any evidence that PD risk gene biomarkers are reflected in sporadic PD? Or as a subgroup of sporadic PD. Could the authors refer to a study where a biomarker such as aSYN seeding is seen in genetic risk, genetic PD and in a proportion of sporadic PD (with significance)? It might be an important point as there is some discussion over whether monogenic PD might be distinct from sporadic PD, risk gene PD.
    • We refer to our response on the next comment by this peer reviewer ("Also line 82, clarify ‘known PD risk genes’ with a few examples."), the addition of PD risk genes and how they affect biology of the disease.
  • Also line 82, clarify ‘known PD risk genes’ with a few examples.
    • we have added a few examples of known PD risk genes, page 2 (last paragraph) and 3 (first paragraph)
  • Line 113. It is correct that viewing tumours as a genetic disease has helped. And considering the heterogeneity of PD, it could help also. But most cases of PD are sporadic and DNA from those patients often not routinely screened like tumour samples are. Perhaps I misunderstood, but are the authors suggesting that we should identify common genetic (risk?) signatures in PD to be able to have better trials and treatments?
    • in oncology, the change from considering pathology to genetics has made a clear impact on the field, with whole genome sequencing changing the game completely. This is the essence of this review, namely that we may learn from genetics analyses to better understand what drives the disease biology in PD leading up to dopaminergic neuronal loss, and current knowledge in PD points to an increasing number of affected genes which may help in finding common pathways, subgroups, mechanisms amenable to change from more biology directed treatment. 

Round 2

Reviewer 2 Report

The manuscript has been improved. I consider that it can be published in this version.